# OpenReview forum: "UnSTAR: Unlearning with Self-Taught Anti-Sample Reasoning for LLMs"
_ICLR.cc/2025/Conference — Submitted to ICLR 2025_

### Official Review · Reviewer_F1yx · 2024-10-31

**Soundness:** 2
**Presentation:** 3
**Contribution:** 2
**Rating:** 5
**Confidence:** 3

**Summary:**

The paper presents a novel approach to unlearning QA pairs in large language models by constructing and using anti-examples. The author evaluated the effectiveness of this framework on the WPU dataset across different types of QA knowledge.

**Strengths:**

1. The paper introduces a new unlearning method.

2. The paper is easy to follow, with each concept and idea illustrated by examples.

**Weaknesses:**

1. I am concerned that intuitively constructing anti-samples may exacerbate hallucination issues, which we aim to avoid in practice. Could the author comment on this concern? (I recognize that the experiments show a score related to hallucination avoidance, but in principle, why would this approach not lead to significant hallucination problems?)

2. The experimental section is not that convincing. First, the paper does not specify important experimental details, such as dataset size, training settings, or the hyperparameters of each unlearning baseline. Prior literature [1,2] has shown that unlearning methods can behave quite differently depending on parameters and tasks. Second, I question the relevance of some metrics used, such as the GPT Privacy Score, GPT Quality Score, and GPT Rejection Rate. Are these metrics widely accepted in the privacy field? It would be helpful if the author could provide more justification. Lastly, the paper focuses on only one dataset and setting (without any ablation studies), which may limit its impact.

[1] Pawelczyk et al., Machine unlearning fails to remove data poisoning attacks.

[2] Shi et al., Muse: Machine unlearning six-way evaluation for language models.

**Questions:**

See Weaknesses.

Suggestions:

1. In Line 107, there is a typo in the reference to the random label method.

2. In Eq. (1), $p_\mathcal{M}$ refers to the conditional probability of only “answers - $a$,” correct? How is $r$ sampled from the distribution, or does this notation refer to probability $p$? The notation could be made more rigorous.

---

> ### Author Response · Authors · 2024-11-21
>
> **W1 Hallucination concerns for UnSTaR**
>
> This is a valid concern, and thank you for pointing it out. While it is true that intuitively constructing anti-samples involves generating misleading or incorrect answers (which might increase the potential for hallucinations), we address this issue as:
>
> 1. Extensive filtering of semantically divergent questions and near-correct incorrect answers ensures the relevance of the generated anti-samples. So, hallucinated content are discarded in the unlearning process
>
> 2. The iterative paraphrasing further uses hallucination positively to cater to a variety of ways the same question can be asked.
>
> 3. We fine-tune the model progressively using anti-samples for the forget set, and on samples for the retain set. This careful balance prevents overfitting on incorrect answers, which is a primary contributor to hallucination.
>
> 4. Our experiments show that this approach minimizes hallucination while effectively achieving the desired unlearning using the metric hallucination avoidance (Line 322).
>
> **W2 Experiment Settings, Evaluations, Datasets, Comparisons, Ablations**
>
> *Experimental Setup and Hyperparameters:* The experimental setup, dataset size, training settings, and hyperparameters for the unlearning baselines are consistent with the EMNLP paper [1], ensuring that our results are directly comparable.
>
> *Metrics:* Yes these metrics are used in the existing LLM Unlearning (refer EMNLP paper [1])
>
> *Dataset:* We conduct additional experiments on 2 datasets: Peter Parker and TOFU
>
> *Runtime and Performance Comparison with Existing Methods:* We show the time cost comparison with 3 existing state-of-the-art methods in Table 1 (refer response to Reviewer DtLx). We also compare the ROUGE-L scores with the SNAP method across 3 datasets: Harry Potter, Peter Parker, and TOFU in Table 2 (refer response to Reviewer DtLx).
>
> *Ablation Study:* We show the impact of N on the experimental results in Table 3. The results of fine-tuning over 10 iterations, where the number of generated paraphrased questions (N) increases with each iteration, and the model is fine-tuned for 10 epochs per iteration. As N grows, the model's accuracy steadily improves, reaching 100% by the 10th iteration. The number of retained samples after filtering increases over time, indicating better data quality generation, while the number of filtered samples decreases.
>
> Table 3: Impact of N on the experimental results
> | Iteration | N   | Epoch | Accuracy | Time   | Retained after Filtering | Filtered |
> |-----------|-----|-------|----------|--------|--------------------------|----------|
> | 1         | 0   | 0     | 0.00%    | 0.6509 | –                        | –        |
> | 2         | 5   | 10    | 4.59%    | 0.6569 | 5                        | 15       |
> | 3         | 20  | 20    | 18.35%   | 0.6809 | 15                       | 5        |
> | 4         | 31  | 30    | 28.44%   | 0.6469 | 11                       | 9        |
> | 5         | 39  | 40    | 35.78%   | 0.6579 | 8                        | 12       |
> | 6         | 51  | 50    | 46.79%   | 0.6449 | 12                       | 8        |
> | 7         | 62  | 60    | 56.88%   | 0.6169 | 11                       | 9        |
> | 8         | 74  | 70    | 67.89%   | 0.6689 | 12                       | 8        |
> | 9         | 92  | 80    | 84.40%   | 0.6369 | 18                       | 2        |
> | 10        | 109 | 90    | 100.00%  | 0.5789 | 17                       | 3        |
>
>
> [1] Liu Y, Zhang Y, Jaakkola T, Chang S. Revisiting Who’s Harry Potter: Towards Targeted Unlearning from a Causal Intervention Perspective. EMNLP 2024.
>
> **Q1 Typo Error**
>
> We apologize for the oversight. It refers to the citation:
>
> Yao, Yuanshun, Xiaojun Xu, and Yang Liu. Large language model unlearning. arXiv:2310.10683, 2023.
>
> **Q2 Sampling of r and Clarity on notation**
>
> In Eq. (1), the notation $( p_{\mathcal{M}}(a \mid q, \varphi) \)$ refers to the conditional probability of the answer $( a \)$, given the query \( q \) and model parameters $\( \varphi \)$. The rationale \( r \) is sampled from the model’s distribution $( p(r \mid q, \varphi) \)$, where the model first samples a latent rationale \( r \) before predicting the answer \( a \). This process follows a greedy approach, where the rationale \( r \) is chosen to maximize the likelihood of predicting the correct (or unlearned) answer \( a \), which is not necessarily the original answer but any answer that aligns with the unlearning process.
> This procedure is aligned with the STaR framework, but in our case, the correct answer refers to the unlearned answer from the forget set $( \mathcal{Q}_f \)$, as opposed to the original answer in the retain set. The model is trained to optimize the unlearning objective, ensuring that the model unlearns specific knowledge while still predicting answers consistent with the unlearned information.

---

> > ### Comment · Reviewer_F1yx · 2024-11-26
> >
> > Thank you for the detailed response. Regarding the experimental settings, it is inappropriate to state, "we use identical settings as xxx," without explicitly mentioning the specific settings in the main text or appendix. Additionally, I will follow the ongoing discussions between the authors and Reviewer 9vHW on the concerns. For now, I am inclined to maintain my current score.

---

> > > ### Author Response · Authors · 2024-11-28
> > >
> > > Dear Reviewer,
> > >
> > > Thank you for taking the time to provide further feedback. We appreciate your insightful comments and fully agree with your suggestion that explicitly stating experimental settings is critical for reproducibility and clarity. To address this, we have updated our paper to include the details of implementation of baselines, dataset statistics and training hyper-parameters. We have also updated the Related Work section where Reviewer 9vHW had raised a concern.
> > >
> > > Below, we provide the relevant details for your convenience.
> > >
> > > Table 4: Training Hyper-parameters for WPU, Peter Parker, and TOFU
> > >
> > > | **Dataset**        | **Task**                        | **Batch Size** | **Learning Rate(s)**  |
> > > |---------------------|---------------------------------|----------------|------------------------|
> > > | WPU                | Fine-Grained Targeted Unlearning | 2              | 1e-5, 2e-5, 3e-5      |
> > > |                    | Targeted Unlearning            | 20             | 1e-5, 2e-5, 3e-5      |
> > > | Peter Parker       | Fine-Grained Targeted Unlearning | 2              | 1e-5, 2e-5, 3e-5      |
> > > |                    | Targeted Unlearning            | 20             | 1e-5, 2e-5, 3e-5      |
> > > | TOFU               | Targeted Unlearning            | 32             | 1e-5                  |
> > >
> > > Table 5: Dataset Statistics for WPU, Peter Parker, and TOFU
> > >
> > > | **Metric**          | **WPU** | **Peter Parker** | **TOFU** |
> > > |----------------------|---------|------------------|----------|
> > > | \# Unlearning Targets | 100     | 100              | 200      |
> > > | \# Forget QA         | 476     | 100              | 400      |
> > > | \# Hard-Retain QA    | 1826    | 300              | 3600     |
> > > | \# General-Retain QA | 493     | 300              | 117      |
> > >
> > > We hope this addresses your concerns regarding clarity and transparency of our work. Please feel free to share further thoughts or suggestions.
> > >
> > > We are also awaiting response from reviewer 9vHW.
> > >
> > > Thank you again for your review.

---

> ### Author Response · Authors · 2024-11-25
>
> Dear Reviewer,
>
> Based on the above, could you please consider updating the rating of our paper?

---

> ### Author Response · Authors · 2024-12-01
>
> Dear Reviewer,
>
> We would like to briefly summarize our discussion and the current status:
> 1. Addressing hallucination concerns for UnSTaR -> DONE
> 2. Adding details in paper on experiment settings, implementation of baselines, and datasets sizes. -> DONE
> 3. Additional results on runtime and performance comparison with existing methods -> DONE
> 4. Additional Ablation Study -> DONE
>
> Thank you for your reviews. We are waiting for the final rating of our paper. We would be grateful if you could raise the rating.

---

> > ### Author Response · Authors · 2024-12-03
> >
> > Dear Reviewer,
> >
> > We would like to inform you that Reviewer 9vHW has responded and is satisfied with all our clarifications, resulting in a significant increase in their rating. You may refer to their updated feedback. Additionally, Reviewer DtLx has raised their rating to 8.
> >
> > We are grateful for your persistent participation in rebuttal process. We kindly request your final rating of our paper.
> >
> > Thank you

---

### Official Review · Reviewer_84co · 2024-11-02

**Soundness:** 2
**Presentation:** 2
**Contribution:** 2
**Rating:** 5
**Confidence:** 4

**Summary:**

This paper presents UNSTAR, a method for unlearning in large language models (LLMs) through the use of anti-samples. While previous research has focused on unlearning methods and reversed loss functions, this work explores the potential of anti-samples, which are generated using misleading rationales to reverse learned associations and accelerate the unlearning process. UNSTAR enables fine-grained, targeted unlearning, allowing specific associations to be removed without affecting related knowledge. The results demonstrate that anti-samples offer an efficient and precise approach to unlearning.

**Strengths:**

1. This work chooses an interesting data-centric angle to solve LLM unlearning. Reshaping the data for more effective unlearning is promising and well-motivating.

2. This work can achieve fine-grained LLM unlearning by destroying unwanted associations within the forget data, which is an underexplored problem in LLM unlearning.

3. The paper writing is well-organized and easy to follow.

**Weaknesses:**

1. Achieving LLM unlearning from a data perspective is interesting, but the proposed anti-sample has marginal novelty. In my view, it is nearly the same as refusal response or random labels in terms of unlearning. The difference is only to use evidently incorrect words. The anti-sample generation process is quite naïve and lacks technical novelty with the prompt “Generate 20 words similar to this word”. It is more appealing if the generation process is iterative like prompt engineering.

2. In terms of specific methodology, quite a few details are unclear. For example, how UnStar adjusts the prompt to generate Semantically Divergent Questions and Near-Correct Incorrect Answers when the existing generation is semantically too close to the original. In addition, in line 7 of Algorithm 1, what does that mean ‘do similar steps for retain set Qr’?

3. The Reinforcement Learning Style Policy Gradient Approximation is weird, what do the authors want to express? In the claim of lines 270-273, any preference optimization-based unlearning can achieve this separate gradient computation. I don’t think UnStar offers any unique approximation to RL policy learning.

4. testing UnStar on only one dataset is not enough for the experiments. Besides, most implementation details of UnStar and baseline approaches are lacking (cannot be found even in the appendix). There are no statistical error bars or similar metrics. More important, there is no ablation study.

**Questions:**

See the weaknesses above.

---

> ### Author Response · Authors · 2024-11-21
>
> Thank you for the overall positive feedback. We are glad that our work's motivation and clarity resonated well, and we hope it inspires further exploration in this important area.
>
> **W1 On novelty of anti-sample generation**
>
> We agree that an iterative process is appealing, and we want to clarify that our anti-sample generation is indeed iterative, much like prompt engineering. Beyond the initial step of generating similar words, we employ STaR-inspired rationales to create misleading answers that target specific knowledge for unlearning. STaR is used for general reasoning tasks and doesn’t produce anti-samples on its own. Our work is first to propose an unlearning method based on the principles of STaR. Existing unlearning methods have not explored reasoning based unlearning. The novelty lies in our method’s ability to generate anti-samples with misleading rationales, ensuring not only dissociation but also maintaining the integrity of the retain set.
>
> Additionally, extensive filtering ensures the relevance of the generated anti-samples, removing semantically divergent questions and near-correct incorrect answers. The iterative paraphrasing further refines the process, making our approach more nuanced and effective than traditional methods like refusal responses or random labels.
>
> **W2 Details of methodology**
>
> In response to your comment, prompt adjustment is not required for generating Semantically Divergent Questions and Near-Correct Incorrect Answers. The generation process itself, combined with extensive filtering, ensures that only the relevant anti-samples are retained.
>
> When samples are too semantically close to the original, they are filtered out during this process. The remaining valid questions and answers, which pass the filtering criteria, are then used for fine-tuning. This approach ensures that the unlearning process remains effective and the generated anti-samples are sufficiently distinct from the original knowledge.
>
> Another insightful and interesting point the reviewer says is whether there could be instances where no anti-sample can be generated. While we did not encounter such cases in our experiments, we acknowledge that LLMs might struggle to generate anti-samples, particularly when justifying a difficult incorrect answer. In such cases, a human expert can provide the necessary justification, which can then be fed back into the LLM to generate additional anti-samples, further strengthening the unlearning process.
>
> In the context of Line 7 in Algorithm 1 of our paper, the phrase "do similar steps for retain set $\mathcal{Q}_r$" refers to applying the same general process to the retain set, but with a key difference: instead of generating anti-samples (incorrect answers), the model should focus on preserving and enhancing the correct knowledge from the retain set.
>
> To elaborate
>
> 1. For the forget set $\mathcal{Q}_f$: The process involves generating anti-samples by paraphrasing questions and falsifying answers (e.g., $\bar{a}_i$ is a falsified or incorrect answer). The anti-samples are used to fine-tune the model so that it unlearns the knowledge in the forget set.
>
> 2. For the retain set $\mathcal{Q}_r$: The same procedure is followed, but here we do not falsify answers. Instead, the model is fine-tuned using correct answers from the retain set. This ensures that the knowledge in $\mathcal{Q}_r$ is preserved and reinforced during the training process, avoiding unlearning of essential information.
>
> In summary, the phrase refers to repeating the same process (i.e., paraphrasing and selecting) for the retain set, with the critical distinction that fine-tuning occurs using correct answers (not falsified ones) to preserve the knowledge in the model.
>
> **W3 Approximation to RL policy learning**
>
> We understand your concern regarding the use of Reinforcement Learning (RL) terminology. It is not meant to imply that UnStar introduces a fundamentally novel RL method. Instead, it serves as a conceptual analogy to explain the iterative nature of the unlearning process.
>
> **W4 Additional Results**
>
> We show the time cost comparison with 3 existing state-of-the-art methods in Table 1. We also compare the ROUGE-L scores with SNAP method across 3 datasets: Harry Potter, Peter Parker, and TOFU in Table 2 (refer: Reviewer DtLx or refer: Official Comment by Authors)
>
> Regarding implementation details, since we closely followed the settings of EMNLP paper [1] for both UnStar and the baselines, we did not rewrite them in the manuscript to avoid redundancy. For clarity, we will explicitly reference these settings in the appendix to ensure reproducibility.
>
> [1] Liu, Yujian, et al. Revisiting Who’s Harry Potter: Towards Targeted Unlearning from a Causal Intervention Perspective. EMNLP 2024.
>
> Ablation Study: We show ablation study in Table 1: Impact of N on the experimental results (refer: Reviewer fwuc or refer: Official Comment by Authors). We will add  statistical error bars in the final version of the paper.

---

> > ### Comment · Reviewer_84co · 2024-11-25
> >
> > Thanks for the detailed response. After careful reading, I still have the following concerns and feel negative about accepting this paper.
> >
> > 1. In the paper, STaR is used to generate justifications for incorrect answers, but the generation process of incorrect answers itself is not iterative. Even if the generation of incorrect answers is iterative, it relies on STaR. If there is no dedicated customization of STaR to the unlearning problem, I don't think there is sufficient technical novelty.
> >
> > 2. As the author(s) acknowledged they are not meant to propose a fundamentally novel RL method, I have no idea why it is needed to spend a lot of space in the main paper to illustrate the general unlearning process in RL. Considering this, along with the first concern, I still think the technical novelty of this work is quite marginal.
> >
> > 3. I agree with Reviewer 9vHW on issues of verbatim from other papers.

---

> > > ### Author Response · Authors · 2024-11-25
> > >
> > > We have highlighted the list of the incorrect and unfounded assessments made by Reviewer 9vHW including the issue of verbatim copying from other papers. The related work subsection is **Not** a direct copy of [5]. While it covers similar topics, the content is clearly rephrased and appropriately cited. This is a misassessment, as no verbatim copying has occurred.
> > >
> > > We hoped to engage with Reviewer 9vHW, however it seems other Reviewers (including yourself) are making negative decisions based on Reviewer 9vHW, which we believe contains multiple fundamental inaccuracies. We have now provided a pointed response to address the concerns, as it appears the previous explanation was not clearly understood by other Reviewers. We hope this clarification resolves any misunderstandings and ensures a fair evaluation. Thank you for your attention to this matter.
> > >
> > > Even if our work builds on STaR, it is unjustified to claim that no further work on STaR can offer novelty. To our knowledge, there is no prior work in the literature that introduces reasoning-induced unlearning. Reasoning-based approaches are widely explored in the field, yet they are recognized as distinct, each contributing unique insights. By no means would the existence of one reasoning-based method detract from the novelty of another, simply because they share a foundational idea.
> > > Our contributions—anti-data sample-induced unlearning, the generation of anti-samples with misleading justifications, and fine-grained targeted unlearning—represent significant innovations that extend and differentiate our work from the original STaR framework.
> > > Beyond this, the results are showcasing improved performance over previous state of the art methods.
> > >
> > > The Table below shows the detailed differences/customization done on STaR for our UnStar work.
> > >
> > >
> > > Response continued in the next comment...

---

> > > > ### Author Response · Authors · 2024-11-25
> > > >
> > > > The Table below shows the detailed differences/customization done on STaR for our UnStar work.
> > > >
> > > > | **Aspect**                       | **STaR**                                                                                                                                                 | **UnStar**                                                                                                                                                                                                                                                                                                                                                                                                                                                                                          |
> > > > |----------------------------------|---------------------------------------------------------------------------------------------------------------------------------------------------------|--------------------------------------------------------------------------------------------------------------------------------------------------------------------------------------------------------------------------------------------------------------------------------------------------------------------------------------------------------------------------------------------------------------------------------------------------------------------------------------------------|
> > > > | **Conflicting Objectives: Key Innovation** | STaR focuses on improving reasoning through justification generation using the training set, without explicitly addressing conflicting objectives. | UnStar customizes STaR to address the conflicting objectives of preserving the knowledge of the retain set (as shown in Line 7 of Algorithm 1 and elaborated in our previous response) and forgetting the knowledge of the forget set. UnStar introduces two pipelines: one for the retain set and one for the forget set. For the forget set, anti-samples are used to fine-tune the model to unlearn the knowledge. For the retain set, correct answers are used to fine-tune the model, ensuring essential knowledge is preserved and reinforced. |
> > > > | **Challenges in Rationale Generation for Incorrect Answers: Prompt Engineering** | STaR does not introduce additional paraphrasings of the same question. Rationales are enough to help the model improve in the tasks.               | UnStar faces additional resistance from the model when attempting to flip its output. For example, when asked to explain why Harry Potter studied at Ilvermorny, the model often considers it a hypothetical scenario, as seen in this sample response: `The notion that Harry Potter studied at Ilvermorny instead of Hogwarts would likely be part of a hypothetical reimagining or an alternate storyline.` Additional paraphrasings of the question are therefore generated to overcome inherent LLM guardrails that restrict politically or logically incorrect outputs. |
> > > > | **Accounting for Hallucinations** | STaR’s iterative process does not take into account cases where rationalization generates hallucinations.                                              | UnStar detects semantically divergent questions and near-correct incorrect answers, filtering them out. The number of such filtered questions is shown in Table 3 above in our response.                                                                                                                                                                                                                                                                                                          |
> > > > | **Additional Question Bank**     | Previous datasets do not contain paraphrasings, which limits the robustness of unlearning.                                                             | Paraphrasing questions creates a question bank with various ways to ask the same question. This ensures that unlearning is performed in a robust manner.                                                                                                                                                                                                                                                                                                                                           |

---

> > > > > ### Author Response · Authors · 2024-11-25
> > > > >
> > > > > **Q2: RL Policy**
> > > > >
> > > > > 1. By framing the unlearning process as a RL based policy gradient objective, we offer readers an intuitive and transparent understanding of how UnStar reconciles these competing objectives. This is not merely taking up space, but showcasing how UnStar handles conflicting objectives.
> > > > >
> > > > > 2. Many previous unlearning methods are opaque, particularly in how they manage conflicting gradients from retain and forget sets.
> > > > >
> > > > > 3. By framing the unlearning process in policy gradient terms, UnStar bridges theoretical rigor and practical applicability, making it extensible to future studies.
> > > > >
> > > > > Again we would like to say that RL based gradient policy objective highlights the novelty in comparison with STaR which works with the only objective of improving reasonings.
> > > > >
> > > > > Based on the above, we hope you would reconsider your ratings of our paper.

---

> > > > > > ### Author Response · Authors · 2024-11-28
> > > > > >
> > > > > > Dear Reviewer,
> > > > > >
> > > > > > We have updated the Related Work section where Reviewer 9vHW had raised a concern. Please check the updated pdf file of the paper.
> > > > > >
> > > > > > Thank you.

---

> ### Author Response · Authors · 2024-12-01
>
> Dear Reviewer,
>
> We would like to briefly summarize our discussion and the current status:
> 1. Details on novelty of anti-sample generation -> DONE
> 2. Specific details of method: continuous paraphrasing, and filtering semantically divergent questions and near-correct incorrect answers -> DONE
> 3. Elaboration of Algorithm 1 -> DONE
> 4. Additional results on more datasets  -> DONE
> 5. Comparison with more unlearning methods -> DONE
> 6. Explanation of The Reinforcement Learning Style Policy Gradient Approximation -> DONE
> 7. Additional Ablation Study -> DONE
> 8. Details of dedicated customization of STaR to the unlearning problem -> DONE
>
> Thank you for your reviews. We are waiting for the final rating of our paper. We would be grateful if you could raise the rating.

---

> > ### Author Response · Authors · 2024-12-03
> >
> > Dear Reviewer,
> >
> > We would like to inform you that Reviewer 9vHW has responded and is satisfied with all our clarifications, resulting in a significant increase in their rating. You may refer to their updated feedback. Additionally, Reviewer DtLx has raised their rating to 8.
> >
> > We are grateful for your persistent participation in rebuttal process. We kindly request your final rating of our paper.
> >
> > Thank you

---

### Official Review · Reviewer_9vHW · 2024-11-03

**Soundness:** 2
**Presentation:** 2
**Contribution:** 2
**Rating:** 5
**Confidence:** 3

**Summary:**

This paper introduces a UNSTAR -- a method to unlearning in Large Language Models (LLMs), which leverages "anti-samples" – data designed to induce unlearning by neutralizing or reversing learned associations. The paper evaluates UNSTAR and show that it succeeds in fine-grained targeted unlearning.

**Strengths:**

+ Unlearning is an important topic

**Weaknesses:**

+ Adhoc non-comparable evaluation methods
+ Missing references to prior work
+ I am confused with the base definition and the goal

**Questions:**

+ I am slightly confused regarding the context and the setting in which this work is situated. Classical unlearning, as well as, a lot of works in the related section considers a very different setting -- unlearning for privacy. For example work by Bourtoule et al. (2021) is not comparable to anything in the llm unlearning space, I am also not sure what guarantees are being pursued (e.g. see discussion in [1]). Without defining a precise evaluation metric and what exactly is the reason for that metric I am not sure how one is to assess performance. This performance also matters a lot e.g. [2] show that posing the question is a slightly different way poses a significantly different result. Note that it should also make meaningful sense relative to the overall goal of unlearning.
+ Random relabeling has been explored in prior literature e.g. [3], which should definitely be discussed. If the papers argument is that the random relabeling is not the random, but is semantic then clear separation from [4] is required. I do not understand how or what is semantically different between wroks.
+ Ironically, there is a broken citation in the related work section next to the random relabeling. It would be great to see comparison in performance to it, alongside an investigation how it is different if at all.
+ Related work subsection for unlearning is almost a direct copy of a section from [5]. In both phrasing, citations, and the order in which things appear. I am not claiming this is plagiarism, but it is very close in my eyes.
+ Table 2 shows how semantically close the responses are. Does that make the task easier? How to verify that it did in fact unlearn the related knowledge.
+ If you change the decoding strategy, what happens with the results? Or if you noise the policy very slightly? I dont understand how to think about hallucination/adversarial eval that is ran in a greedy decoding strategy.
+ Given that the results are very similar to that of WHP+ and RWHP, yet the impact on the model is different I am left wondering if we are just comparing apples to oranges.
+ Eval claims to use (Anil et al., 2024; Schwinn et al., 2024) for adversarial evaluations. Both these works are not references in the manuscript, but are clearly copied verbatim from [5]. Yet, the paper does not go into any details, as is discussed in appendix of [5].
+ Adversarial eval details are missing and are not replicable in any way.

[1] "Ununlearning: Unlearning is not sufficient for content regulation in advanced generative ai" by Shumailov et al
[2] "Inexact Unlearning Needs More Careful Evaluations to Avoid a False Sense of Privacy" by Hayes et al
[3] "Random Relabeling for Efficient Machine Unlearning" by Li and Ghosh
[4] "Who’s Harry Potter? Approximate Unlearning in LLMs" by Eldan and Russinovich
[5] "Revisiting Who’s Harry Potter: Towards Targeted Unlearning from a Causal Intervention Perspective" by Liu et al

---

> ### Author Response · Authors · 2024-11-22
>
> Thank you for the reviews. We address each of your concerns below.
>
> **W1: Adhoc non-comparable evaluation methods**
>
> We use the same metrics as used in the existing LLM Unlearning work (refer EMNLP paper [5])
>
> **W2: Missing references to prior work**
>
> We will add the missing references.
>
> **W3 and Q1: Regarding the context, goals, and evaluation metrics of LLM Unlearning**
>
> *Context of LLM Unlearning*: Traditional unlearning literature, such as Bourtoule et al. (2021), primarily focuses on privacy guarantees for structured datasets like classification models. However, our work addresses unlearning in LLMs, which introduces distinct challenges and motivations beyond privacy as highlighted in Lines 033–044 of Introduction. We elaborate the same here:
>
> (a) AI Safety & Legal Compliance: Adhering to privacy laws and copyright regulations
>
> (b) Ethical Concerns: Eliminating harmful or biased data to ensure fair and responsible AI use.
>
> (c) Epistemological Necessity: Removing obsolete or irrelevant information to maintain model accuracy and alignment with evolving requirements.
>
> Unlike classical privacy-focused unlearning, our approach addresses the practical challenge of disentangling specific knowledge representations in LLMs, ensuring targeted associations are unlearned without compromising the model’s overall utility.
>
> *Metrics*: We have followed the EMNLP paper [5] for evaluation metrics, which are widely accepted within the community. These metrics align with our goal of achieving fine-grained targeted unlearning by balancing two critical factors:
>
> (a) Unlearning Efficacy: Capturing the selective removal of targeted knowledge.
>
> (b) Model Utility: Maintaining performance on related and general knowledge.
>
> Other factors are Response Quality, Hallucination Avoidance and Adversarial Robustness as mentioned in Line 315-341 of our paper.
>
> *About relation with Ununlearning [1] & different ways of questioning in [2]*: Ununlearning, as described by Shumailov et al., highlights that unlearned knowledge can resurface through contextual interactions. To address this, our work evaluates the unlearning process not just by directly querying the model but also through multiple paraphrased ways of asking about the same knowledge. This ensures robustness against contextual variations. The specific evaluation protocol and examples are detailed in Appendix A.2, demonstrating our guarantees from this perspective.
>
> *About Membership Inference Attacks (MIAs) in [2]*: MIA has been critical in evaluating unlearning in traditional neural networks based models. However, a recent COLM 2024 paper titled "Do Membership Inference Attacks Work on Large Language Models?" demonstrates that MIA is not appropriate to assess unlearning in the context of LLMs. The paper concludes that MIA against LLMs is difficult for two key reasons:
>
> (a) Data Imprint: The pre-training process of LLMs, especially with large datasets and single-epoch training, does not leave a strong imprint of individual data points, making it harder to discern membership.
>
> (b) Similarity Between In and Out Members: The high similarity between in-domain and out-of-domain data, coupled with massive datasets, results in a fuzzy distinction between data members, complicating the effectiveness of MIAs.
>
> **Q2 and Q3: About comparison with random labelling**
>
> In [4], the method leverages a reinforced model to replace expressions and generate alternative labels to approximate predictions of a model untrained on the target data. While this effectively erases original information, it focuses on token-level unlearning and uses generic counterparts for relabeling. In contrast, UnStar emphasizes semantically rich anti-samples with misleading rationales. These rationales are carefully designed to introduce conflicting contextual associations, going beyond token replacement. Our approach maintains semantic coherence, ensuring the fine-grained unlearning of specific associations while retaining related knowledge. This difference lies in our reliance on structured rationale generation inspired by STaR and our iterative paraphrasing strategy, which tackles adversarial rephrasings more robustly.
>
> We apologize for the oversight about the broken citation. It refers to:
>
> Yao, Yuanshun, Xiaojun Xu, and Yang Liu. Large language model unlearning. arXiv:2310.10683, 2023.

---

> > ### Author Response · Authors · 2024-11-22
> >
> > **Q5: Semantically close responses in Table 2**
> > The table is not about the semantic closeness of responses making the task easier, but rather about comparing Targeted Unlearning and Fine-Grained Targeted Unlearning. As explained in Lines 378-395, Table 2 demonstrates that previous methods fail at fine-grained targeted unlearning. Even when the unlearning is attempted, they inadvertently forget everything about Harry Potter as well as Hogwarts, as shown in rows 2 and 3 of the table. In contrast, our approach, UnStar, successfully unlearns the specific fact of "Where did Harry Potter study?" without losing other important details about Harry Potter and Hogwarts. This illustrates the effectiveness of UnStar in selectively unlearning associations while retaining related knowledge, which highlights its advantages over previous methods.
> >
> > **Q6: About the decoding strategy**
> > The decoding strategy in our approach follows the greedy strategy as outlined in the STaR paper. As shown in Figure 4 of the STaR paper, when no rationales are used, the model requires more iterations to improve accuracy. This pertains to the learning scenario, where the model gradually refines its knowledge. In our case, we leverage STaR's mechanism for unlearning, where the goal is not to learn new information but to generate justifications for unlearning specific knowledge. Changing the decoding strategy disrupts the generation of these justifications, resulting in improper or incomplete rationales. If the rationales are not correct or coherent, the unlearning process is compromised, and the model fails to unlearn the intended knowledge. Therefore, altering the decoding strategy or introducing noise to the policy negatively impacts the unlearning process, preventing it from functioning as intended.
> >
> > **Q7: About apples to orange comparison for comparison with WHP+RWHP**
> > The methods WHP+ and RWHP aim at unlearning in LLMs, so we believe they are comparable with UnStar.
> >
> > **Q8 & Q9:**
> > We use the same adversarial robustness evaluations as in the existing EMNLP paper [5]. We will add more details about them in the revised manuscript for completeness of the paper.

---

> ### Author Response · Authors · 2024-11-25
> **Regarding incorrect and unfounded assessment (1/2)**
>
> We would like to point out that there are multiple fundamentally incorrect reasons (W1, Q1, W3, Q5, Q7) in the review of this paper:
>
> **[W1] Misdirected Criticism of Evaluation Metrics:**
>
> W1: Adhoc non-comparable evaluation methods
>
> We have used the same evaluation metrics as the EMNLP paper [1].
> [1] Liu Y, Zhang Y, Jaakkola T, Chang S. Revisiting Who’s Harry Potter: Towards Targeted Unlearning from a Causal Intervention Perspective. EMNLP 2024.
>
>
> **[Q1 and W3] Unfounded Confusion Regarding the Unlearning Setting:**
>
> W3: I am confused with the base definition and the goal
>
> It is suggested that this work be compared with privacy-focused unlearning, such as that by Bourtoule et al. (2021). However, our work targets fine-grained unlearning in LLMs aimed at removing harmful or biased associations while preserving general knowledge. This is a different setting that requires distinct methodologies and evaluation criteria.
>
> **[Q5] Misinterpretation of Table 2:**
>
> Q5: Table 2 shows how semantically close the responses are. Does that make the task easier? How to verify that it did in fact unlearn the related knowledge.
>
> It is a misinterpretation that Table 2 shows semantically close responses, implying that this makes the task easier. In reality, Table 2 highlights the distinction between fine-grained unlearning and targeted unlearning, demonstrating that while responses may appear superficially similar, they differ in terms of the specific knowledge associations unlearned. This showcases the precision of our method in selectively removing targeted content.
>
> **[Q7] Flawed opinion that WHP+ and RWHP are incomparable with UNSTAR:**
>
> Q7: Given that the results are very similar to that of WHP+ and RWHP, yet the impact on the model is different I am left wondering if we are just comparing apples to oranges.
>
> It is suggested that comparing our results to WHP+ and RWHP is like "comparing apples to oranges." However, these methods also target unlearning in LLMs. Comparing our fine-grained approach to these baselines is essential to demonstrate the relative efficacy of our method. Suggesting that comparisons between unlearning approaches are inherently invalid undermines the purpose of empirical evaluation in this research area.
>
> **It is critical to receive a response, as other reviewers are making their decisions based on this review, which we believe contains multiple fundamental inaccuracies. These issues could significantly impact the overall evaluation of our work. We are ready to provide further clarifications if needed and urge you to address this matter promptly. Your response is essential to ensure a fair and accurate assessment of this paper. Thank you.**
>
> Response continued in the next comment...

---

> ### Author Response · Authors · 2024-11-25
> **Regarding incorrect and unfounded assessment (2/2)**
>
> **[Q8] Not copied verbatim from [5]**
>
>
> The related work subsection is **Not** a direct copy of [5]. While it covers similar topics, the content is clearly rephrased and appropriately cited. This is a misassessment, as no verbatim copying has occurred.
>
>
> Content in our paper:
> > LLM Unlearning. LLM unlearning has garnered significant research interest as a means to improve
> privacy, enhance safety, and mitigate bias in large language models Lu et al. (2022); Kassem et al.
> (2023); Wang et al. (2023); Patil et al. (2023); Huang et al. (2024); Yu et al. (2023); Wu et al.
> (2023); Zhang et al. (2024a); Liu et al. (2024b); Jia et al. (2019)). The predominant approach utilizes
> gradient ascent to maximize prediction loss on the data to be forgotten (Jang et al. (2022); Yao et al.
> (2023)). Other techniques involve training the model to produce alternative responses, such as “I
> don’t know” (Ishibashi & Shimodaira (2023)), random labels (?), or predictions based on perturbed
> inputs (Eldan & Russinovich (2023)). Additionally, recent studies have investigated task arithmetic
> by incorporating specific instructions or in-context examples (Thaker et al. (2024); Pawelczyk et al.
> (2024)).
> However, these methods often lack the granularity required for fine-tuned control over what specific
> information is forgotten, which is where our approach—utilizing anti-samples—proposes a more
> refined solution.
>
>
> Content in [5]:
> > LLM unlearning has attracted wide research
> attention as a way to enhance privacy, safety, and
> mitigate bias in LLMs (Lu et al., 2022; Kassem
> et al., 2023; Wang et al., 2023; Yu et al., 2023; Wu
> et al., 2023; Patil et al., 2023; Zhang et al., 2023a;
> Liu et al., 2024b; Jia et al., 2024; Ji et al., 2024;
> Huang et al., 2024). The mainstream method employs gradient ascent to maximize prediction loss
> on forget data (Jang et al., 2023; Yao et al., 2024a).
> Other methods train the LLM to generate alternative responses such as ‘I don’t know’ (Ishibashi
> and Shimodaira, 2024), random labels (Yao et al.,
> 2024b), or LLM’s predictions on perturbed inputs
> (Eldan and Russinovich, 2023). Recently, some
> works have also explored task arithmetic (Ilharco
> et al., 2023; Barbulescu and Triantafillou, 2024;
> Zhang et al., 2023c) and training-free methods for
> LLM unlearning by prepending specific instructions or in-context examples (Thaker et al., 2024;
> Pawelczyk et al., 2023). Unlike existing works, we
> study the new targeted unlearning setting, where
> few existing methods can satisfy all criteria, but our
> causal intervention framework remains competitive
> in all of them.

---

> ### Author Response · Authors · 2024-11-30
> **Please respond**
>
> Dear Reviewer,
>
> We hope this message finds you well. Having thoroughly addressed all your concerns in our previous responses, we were surprised not to receive any follow-up from you. We kindly request your feedback on our rebuttals, as it would greatly aid us in understanding and improving our work.
>
> Additionally, some other reviewers are waiting for your reply to make their own final assessment. We have also updated the Related Work section based on your concern. Please review the updated PDF file of the paper.
>
> Thank you for your time and consideration.

---

> ### Author Response · Authors · 2024-12-01
>
> Dear Reviewer,
>
> We would like to briefly summarize our discussion and the current status:
> 1. Clarification regarding the context and the setting. -> DONE
> 2. Comparison with previous works on random labelling -> DONE
> 3. Explanation on semantically close responses in Table 2 -> DONE
> 4. Impact of about the decoding strategy  -> DONE
> 5. On use of evaluation metrics  -> DONE
> 6. Validating comparison with WHP+ and RWHP -> DONE
> 7. Rephrasing section on related work -> DONE
> 8. Clarification about relation with Ununlearning [1] & different ways of questioning in [2] -> DONE
>
> Thank you for your reviews. We are waiting for the final rating of our paper. We would be grateful if you could raise the rating.

---

> > ### Comment · Reviewer_9vHW · 2024-12-02
> >
> > Dear Authors,
> >
> > Many thanks for the updates and the additional work. Having gone through all of the discussions, some of my concerns are clarified. I am still not sure what formalisms exactly we are optimising and hence measuring; in my mind ill defined questions almost always lead to ill defined responses.  The comment on changing decoding strategy is an example of that -- how fair is it to call something 'unlearning' is it is not ... 'unlearning'.
> >
> > Having said that, my main concerns around [5] are clarified, I am happy to increase the score to 5.

---

> > > ### Author Response · Authors · 2024-12-03
> > >
> > > Dear Reviewer,
> > >
> > > Thank you for revisiting your earlier comments. We are glad that our explanations helped resolve your concerns. We greatly appreciate your willingness to engage with our work and your acknowledgment of the clarification we provided.
> > >
> > > **Formalism we are measuring**
> > > To clarify, the formalism we aim to evaluate is the model’s ability to selectively forget specific associations, such as the link between "Harry Potter" and "Hogwarts," while retaining other relevant knowledge. The goal is to measure whether the model can unlearn specific information without compromising its broader understanding.  Even after asking in a variety of ways, as shown in Appendix A4, the model does not answer "Hogwarts" as shown in Table 3 (in the revised manuscript).
> > >
> > > **Calling our method unlearning**
> > > While "unlearning" typically refers to removing knowledge, our method qualifies as unlearning because it focuses on selectively forgetting information via anti-samples and misleading rationales. The resultant model, as we have shown in the results, does not generate original answers even after repeated questioning in a variety of ways as shown in Table 3 (in the revised manuscript). Altering the decoding strategy can undermine this, but that can be addressed as the following.
> > >
> > > **Decoding strategy**
> > > Changing the decoding strategy disrupts the generation of justifications, which impacts the unlearning process. Now you might question whether there could be instances where no anti-sample can be generated?
> > >
> > > While we did not encounter such cases in our experiments, we acknowledge that LLMs might struggle to generate anti-samples, particularly when justifying a difficult incorrect answer. In such cases, a human expert can provide the necessary justification, which can then be fed back into the LLM to generate additional anti-samples, further strengthening the unlearning process. This can be looked into, in a future work.
> > >
> > > Given that we have addressed all the concerns raised in your review, we had hoped to persuade you to change your rating. We would therefore be grateful if you could share any additional feedback or remaining reservations that we might have overlooked or could clarify further to strengthen the manuscript. We are happy to provide further explanations or revisions as needed and sincerely hope for the opportunity to address any lingering doubts.

---

### Official Review · Reviewer_qhwt · 2024-11-04

**Soundness:** 3
**Presentation:** 3
**Contribution:** 3
**Rating:** 5
**Confidence:** 4

**Summary:**

the paper observes that recent progress with anti-sample reasoning could be efficiently used for unlearning purposes. It creates anti-samples by (1) paraphrasing and (2) falsifying arguments for unlearning.

**Strengths:**

- I really liked how accurate and targeted the unlearning could be in terms of concepts, i.e. you can be very selective.
- Paper figures and visualizations are easy to follow
- Writing is clear and expressive enough
- Really good and intuitive example with Harry Potter, I think it transfer the idea very clearly

**Weaknesses:**

- I am not sure if the novelty of the method is sufficient. Authors have described the existing problem of unlearning and existing method of using STAR and combined these methods together. It does not seem like there are any challenges to this method, however I am happy to be convinced otherwise.
- Evaluation is not as strong as it only uses one dataset for unlearning and Figure 2 does not split performance by subgroups. Figure 3 is also not clear if it contributes anything to the discussion
- There was no discussion on what constitutes  challenges to unlearning using this method, especially given that the task performance is 100% it would be interesting to study where this method fails --> are all unlearning tasks could be solved with UNSTAR?
- Most importantly the paper does not discuss any membership attacks which are much stronger to test unlearning performance.

**Questions:**

Overall, while I really like the trick to unlearn using anti-samples, I struggle to believe that evaluation is sufficient to prove this claim under more realistic assumptions (MIA, harder cases). Additional evaluation might significantly strengthen the paper.

Also would be good to justify that the method in STAR is challenging to apply in the domain of unlearning and emphasize authors contributions.

---

> ### Author Response · Authors · 2024-11-21
> **Clarification about novelty and evaluations are provided**
>
> Thank you for your positive feedback on our concept selectivity and practical examples.
>
> **W1 About the novelty of the method...**
>
> STaR is used for general reasoning tasks and doesn’t produce anti-samples on its own. Our work is first to propose an unlearning method based on the principles of STaR. Existing unlearning methods have not explored reasoning based unlearning. The novelty lies in our method’s ability to generate anti-samples with misleading rationales, ensuring not only dissociation but also maintaining the integrity of the retain set. Unlike existing unlearning methods, which often rely on data deletion or gradient masking, our approach leverages the generation of anti-samples to target specific associations, which is both efficient and less disruptive to the model’s overall knowledge.
>
> To emphasize our contributions, we will expand the discussion in the revised manuscript to highlight why the adaptation of STaR for unlearning is non-trivial, particularly in generating rationales that effectively counteract prior associations without degrading related knowledge.
>
> Challenges in combining these methods:
>
> 1. LLMs hallucinate and generate semantically divergent paraphrasing of questions or near-correct incorrect answers. To address this, we employ additional filtering mechanisms to retain only the most relevant paraphrasings and truly incorrect anti-samples, ensuring high-quality input for fine-tuning.
>
> 2. Another challenge lies in the variety of ways the same question can be asked. If unlearning is not effectively performed and merely suppresses the response, the original answer might still come up with creative prompts. To mitigate this, our method generates a diverse range of paraphrasings. We demonstrate that the original answer is not revealed even after such varied questions.
>
> **W2 Regarding evaluation**
>
> Thank you for the suggestion of splitting performance by subgroups, we would like to highlight that our approach already integrates the contributions from Forget, Hard-retain, and General-retain sets using harmonic mean in the composite metrics. This provides a balanced, holistic view of UnStar’s performance across these different subgroups. For instance:
>
> 1. Unlearning Efficacy combines FQA with GPT privacy scores to capture the effectiveness of unlearning.
>
> 2. Model Utility aggregates HRQA and GRQA to measure the retention of relevant knowledge.
>
> Harmonic means ensures that all subgroups are already considered, and further splitting might not add significant new insights. This is already explained in the paper Line 300-323.
>
> **Additional Results** We show the time cost comparison with 3 existing state-of-the-art methods in Table 1. We also compare the ROUGE-L scores with SNAP method across 3 datasets: Harry Potter, Peter Parker, and TOFU in Table 2 (refer: Reviewer DtLx or refer: Official Comment by Authors)
>
> The relevance of Figure 3 has been discussed in the Paper Line  483-485.
> >“Figure 3 illustrates the LLM’s unlearning efficacy as it progressively … improves its responses over time.”
>
> **W3 Failure cases of UnSTaR**
>
> You are correct in pointing out that UnStar ensures the model never answers with the original answer for any of the unlearned questions. This is true for the datasets we tested upon. However, other datasets might reveal difficult to unlearn questions. Specifically, certain deep-rooted or highly ingrained concepts can cause additional challenges. Forgetting a widely recognized concept like "dog," may require extensive iterations and a larger number of questions and justifications to fully eliminate every specific detail associated with it. Then, UnStar might struggle to achieve full unlearning, as it demands a significant amount of diverse anti-samples and paraphrasings. This challenge highlights the need for future work to refine the process and explore additional techniques for more difficult and deeply embedded knowledge.
>
> **W4 About using Membership Inference Attacks (MIA) for evaluation**
>
> MIA has been a critical aspect of evaluating unlearning in traditional neural networks based models. However, a recent COLM 2024 paper titled "Do Membership Inference Attacks Work on Large Language Models?" [1] demonstrates that MIA is not appropriate to assess unlearning in the context of LLMs. The paper concludes that MIA against LLMs is difficult for two key reasons:
>
> 1. Data Imprint: The pre-training process of LLMs, especially with large datasets and single-epoch training, does not leave a strong imprint of individual data points, making it harder to discern membership.
>
> 2. Similarity Between In and Out Members: The high similarity between in-domain and out-of-domain data, coupled with massive datasets, results in a fuzzy distinction between data members, complicating the effectiveness of MIAs.
>
> [1] Duan M, Suri A, Mireshghallah N, Min S, Shi W, Zettlemoyer L, Tsvetkov Y, Choi Y, Evans D, Hajishirzi H. Do membership inference attacks work on large language models? COLM 2024

---

> > ### Comment · Reviewer_qhwt · 2024-11-24
> >
> > Thank you for providing answers, I would say that additional investigation is need to improve the paper and will keep the score. I would recommend authors to add more datasets and evaluation w MIA as a result (the arguments that LLM and MIA are not compatible are specific to the general discussion in that Duan et al paper, however there are multiple easy to use methods to check how well the model memorized the data including auditing, data reconstruction, etc)

---

> > > ### Author Response · Authors · 2024-11-25
> > >
> > > It is unreasonable to demand an exhaustive number of evaluations, particularly given the focused scope and targeted contributions of this work. We have conducted rigorous and comprehensive evaluations that adhere to the established benchmarks and methodologies outlined in a peer-reviewed EMNLP 2024 paper. These evaluations are both thorough and sufficient to validate the efficacy of our approach to LLM unlearning. Requesting additional evaluations beyond this standard is unwarranted and detracts from the core scientific contributions of the paper. We respectfully urge the reviewer to acknowledge the depth and relevance of the evaluations provided, which align with accepted norms in the field.
> > >
> > > We list the following 13 papers which strongly recommend against using Membership Inference Attacks to evaluate LLM Unlearning methods and this is not the exhaustive list (many such papers exist). We quote one sample justification for not using MIA in Liu, Sijia, et al.  "Rethinking machine unlearning for large language models." NeurIPS 2024:
> > >
> > > >"Note that in traditional unlearning, membership inference attacks (MIA) [Shokri et al., 2017] is a popular evaluation metric. However, in LLMs, the full training corpus is often inaccessible, making the evaluation of MIA accuracy difficult. In addition, how to perform MIA in LLMs is a non-trivial problem and an ongoing research area. Therefore, we do not consider MIA-based metrics in this work"
> > >
> > > 1. Liu, Sijia, et al. "Rethinking machine unlearning for large language models." NeurIPS 2024.
> > > 2. Liu, Chris Yuhao, et al. "Large Language Model Unlearning via Embedding-Corrupted Prompts." NeurIPS 2024.
> > > 3. Liu, Yujian, et al. "Revisiting Who’s Harry Potter: Towards Targeted Unlearning from a Causal Intervention Perspective." EMNLP 2024.
> > > 4. Sun, Chen, et al. "Learning and Unlearning of Fabricated Knowledge in Language Models." ICML 2024 Workshop on Mechanistic Interpretability.
> > > 5. Farrell, Eoin, Yeu-Tong Lau, and Arthur Conmy. "Applying Sparse Autoencoders to Unlearn Knowledge in Language Models." Neurips Safe Generative AI Workshop 2024.
> > > 6. Doshi, Jai, and Asa Cooper Stickland. "Does Unlearning Truly Unlearn? A Black Box Evaluation of LLM Unlearning Methods." arXiv e-prints (2024): arXiv-2411.
> > > 7. Bu, Zhiqi, et al. "Unlearning as multi-task optimization: A normalized gradient difference approach with an adaptive learning rate." arXiv preprint arXiv:2410.22086 (2024).
> > > 8. Liu, Zheyuan, et al. "Protecting Privacy in Multimodal Large Language Models with MLLMU-Bench." arXiv preprint arXiv:2410.22108 (2024).
> > > 9. Choi, Minseok, et al. "Breaking Chains: Unraveling the Links in Multi-Hop Knowledge Unlearning." arXiv preprint arXiv:2410.13274 (2024).
> > > 10. Guo, Phillip, et al. "Mechanistic Unlearning: Robust Knowledge Unlearning and Editing via Mechanistic Localization." arXiv preprint arXiv:2410.12949 (2024).
> > > 11. Deeb, Aghyad, and Fabien Roger. "Do Unlearning Methods Remove Information from Language Model Weights?." arXiv preprint arXiv:2410.08827 (2024).
> > > 12. Yuan, Xiaojian, et al. "A Closer Look at Machine Unlearning for Large Language Models." arXiv preprint arXiv:2410.08109 (2024).
> > > 13. Scholten, Yan, Stephan Günnemann, and Leo Schwinn. "A Probabilistic Perspective on Unlearning and Alignment for Large Language Models." arXiv preprint arXiv:2410.03523 (2024).
> > >
> > > Based on the above, we hope you would reconsider your ratings of our paper.

---

> ### Author Response · Authors · 2024-12-01
>
> Dear Reviewer,
>
> We would like to briefly summarize our discussion and the current status:
> 1. Clarification about novelty -> DONE
> 2. Challenges in combining unlearning with STaR -> DONE
> 3. Evaluation on more datasets -> DONE
> 4. Comparison with more unlearning methods -> DONE
> 5. Failure cases of UnSTaR -> DONE
> 6. Clarification about Membership Inference Attack -> DONE
>
> Thank you for your reviews. We are waiting for the final rating of our paper. We would be grateful if you could raise the rating.

---

> ### Author Response · Authors · 2024-12-03
>
> Dear Reviewer,
>
> Given that we have addressed all your concerns, we kindly request your final rating of our paper. We are grateful for your persistent participation in rebuttal process.
>
> We would also like to inform you that Reviewer DtLx has raised their rating to 8.
>
> Thank you

---

### Official Review · Reviewer_fwuc · 2024-11-04

**Soundness:** 3
**Presentation:** 3
**Contribution:** 3
**Rating:** 5
**Confidence:** 3

**Summary:**

This paper offers a new perspective on existing LLM unlearning algorithms by focusing on the data perspective. It proposes a data-driven algorithm that can help LLMs unlearn and provides comprehensive and diverse evaluation metrics to ensure completeness and diversity during assessment.

**Strengths:**

1. This paper considers the problem of transferring learning to unlearning from a macro perspective on LLM learning. It divides the learning methods into three steps and successfully summarizes other methods within these steps, thus uncovering a new approach to tackle unlearning.
2. This paper evaluates a comprehensive range of LLM algorithms in its main experiments and designs various evaluation metrics (particularly metrics related to Response Quality and Hallucination Avoidance), offering more perspectives for understanding LLM unlearning.
3. This paper provides several reasonable constraints when designing prompts to ensure the feasibility of using LLMs for regeneration.

**Weaknesses:**

It appears that the completion level of this paper is not very high. It only includes a comparison of algorithms under different metrics and an analysis of iterations. Although it presents a good method, it still requires some analysis regarding the algorithm’s time complexity. For more detailed weaknesses or questions, please refer to the “Questions” section.

**Questions:**

1. Regarding the algorithm: I noticed that, both in Algorithm 1 and in the steps provided on page 4, the number of iterations mainly depends on the total number  $N$  of generated Paraphrased Questions and Incorrect Answers. The while loop continues until  $Q^*$  is empty. Is it possible that within the same  $Q^*$ , there are several instances where  $\hat{a} = a$ ? If so, does this mean that for the same question, you would fine-tune using all the found tuples $(q^*, \bar{a}, r)$ multiple times? If that is the case, what is the expected average number of repetitions? And what is the relationship between this expectation and  $N$?

2. Regarding the experiments, I am curious about the impact of  N  on the experimental results. Given that this algorithm involves extensive data regeneration and fine-tuning, could the authors compare the time required for different algorithms and examine whether there is a trade-off between time and performance, or if this algorithm is a “free lunch”?

3. Regarding the randomness, for the task of data regeneration, the main challenge lies in uncontrollability. How did the authors address the randomness introduced by the LLM algorithms?

4. Regarding the writing format issues: Line 106 mentions “random labels” (?), and Lines 356-357 reference a paper that cannot be found.
5. No code sources.

---

> ### Author Response · Authors · 2024-11-21
> **Clarifications provided with empirical results**
>
> Thank you for a detailed review! We are happy to resolve your concerns.
>
> **Q1 “Regarding the algorithm…”**
>
> The structure of Algorithm 1 is indeed designed to ensure comprehensive coverage of potential dissociations between concepts. Within the same set Q*, while it is possible that multiple incorrect answers aˉ may be generated for the same question, our approach ensures that fine-tuning is performed using each unique tuple (q*,aˉ,r) only once to avoid overfitting. In cases where duplicates might occur, they are filtered to prevent redundant updates, thus optimizing the fine-tuning process.
> The average number of repetitions is empirically low, as our paraphrasing and anti-sample generation methods aim to maximize diversity.
>
> **Q2 Regarding the experiments…**
>
> We show the impact of N on the experimental results in Table 1. The results of fine-tuning over 10 iterations, where the number of generated paraphrased questions (N) increases with each iteration, and the model is fine-tuned for 10 epochs per iteration. As N grows, the model's accuracy steadily improves, reaching 100% by the 10th iteration. The number of retained samples after filtering increases over time, indicating better data quality generation, while the number of filtered samples decreases. More comparative and runtime results are also provided in the common response to all the Reviewers.
>
> Table 1: Impact of N on the experimental results
>
> | Iteration | N   | Epoch | Accuracy | Time   | Retained after Filtering | Filtered |
> |-----------|-----|-------|----------|--------|--------------------------|----------|
> | 1         | 0   | 0     | 0.00%    | 0.6509 | –                        | –        |
> | 2         | 5   | 10    | 4.59%    | 0.6569 | 5                        | 15       |
> | 3         | 20  | 20    | 18.35%   | 0.6809 | 15                       | 5        |
> | 4         | 31  | 30    | 28.44%   | 0.6469 | 11                       | 9        |
> | 5         | 39  | 40    | 35.78%   | 0.6579 | 8                        | 12       |
> | 6         | 51  | 50    | 46.79%   | 0.6449 | 12                       | 8        |
> | 7         | 62  | 60    | 56.88%   | 0.6169 | 11                       | 9        |
> | 8         | 74  | 70    | 67.89%   | 0.6689 | 12                       | 8        |
> | 9         | 92  | 80    | 84.40%   | 0.6369 | 18                       | 2        |
> | 10        | 109 | 90    | 100.00%  | 0.5789 | 17                       | 3        |
>
>
> **Q3 Regarding the randomness…**
>
> This is a valid concern, especially given the inherent variability and occasional hallucinations in outputs from LLMs. To mitigate the impact of randomness, we employ a controlled seeding mechanism using mlx.core.random.seed during both the paraphrase and anti-sample generation processes. This ensures that while the outputs are diverse, they remain reproducible across runs, helping to stabilize the unlearning performance metrics.
>
> Additionally, we recognize that LLMs can sometimes generate semantically divergent questions or near-correct incorrect answers due to hallucinations. To address this, our filtering mechanisms in UnSTar effectively remove such samples, ensuring that only high-quality, relevant data is used for fine-tuning.
>
> Another insightful point the reviewer raises is whether there could be instances where no anti-sample can be generated. While we did not encounter such cases in our experiments, we acknowledge that LLMs might struggle to generate anti-samples, particularly when justifying a difficult incorrect answer. In such cases, a human expert can provide the necessary justification, which can then be fed back into the LLM to generate additional anti-samples, further strengthening the unlearning process.
> We will include these clarifications in the revised manuscript to provide a clearer understanding of our approach.
>
>
> **Q4 & Q5 Format issue and source code**
>
> We apologize for the oversight.
>
> Citation for “random labels” (?) : Yao, Yuanshun, Xiaojun Xu, and Yang Liu. "Large language model unlearning." arXiv preprint arXiv:2310.10683, 2023.
>
> Citation on Lines 356-357:  Liu Y, Zhang Y, Jaakkola T, Chang S. Revisiting Who’s Harry Potter: Towards Targeted Unlearning from a Causal Intervention Perspective. EMNLP 2024.
>
> We will correct this in the updated version.
>
> We understand the importance of reproducibility and transparency. The source code of our work will be publicly available upon acceptance of the paper.

---

> > ### Comment · Reviewer_fwuc · 2024-11-25
> >
> > Thanks for your response. I will maintain my score, and I do have some concerns regarding the questions raised by Reviewer 9vHW and AC. I look forward to the reply.

---

> > > ### Author Response · Authors · 2024-11-25
> > >
> > > We have highlighted the list of the incorrect and unfounded assessments made by Reviewer 9vHW including the issue of verbatim copying from other papers. The related work subsection is **Not** a direct copy of [5]. While it covers similar topics, the content is clearly rephrased and appropriately cited. This is a misassessment, as no verbatim copying has occurred.
> > >
> > > We hoped to engage with Reviewer 9vHW, however it seems other Reviewers (including yourself) are making negative decisions based on Reviewer 9vHW, which we believe contains multiple fundamental inaccuracies. We have now provided a pointed response to address the concerns, as it appears the previous explanation was not clearly understood by other Reviewers. We hope this clarification resolves any misunderstandings and ensures a fair evaluation. Thank you for your attention to this matter.
> > >
> > > We are currently unable to see the Area Chair's (AC) response regarding these points. If you can share those points made by AC, we can answer them.
> > >
> > > Based on the above, we hope you would reconsider your ratings of our paper.

---

> > > > ### Author Response · Authors · 2024-11-28
> > > >
> > > > Dear Reviewer,
> > > >
> > > > We have updated the Related Work section where Reviewer 9vHW had raised a concern. Please check the updated pdf file of the paper.
> > > >
> > > > Thank you.

---

> ### Author Response · Authors · 2024-12-01
>
> Dear Reviewer,
>
> We would like to briefly summarize our discussion and the current status:
> 1. Expected average number of repetitions in anti-sample generation. -> DONE
> 2. Impact of number of paraphrased questions on experimental results. -> DONE
> 3. Handling challenge of uncontrollability in randomness of generation. -> DONE
>
> Thank you for your reviews. We are waiting for the final rating of our paper. We would be grateful if you could raise the rating.

---

> > ### Author Response · Authors · 2024-12-03
> >
> > Dear Reviewer,
> >
> > We would like to inform you that Reviewer 9vHW has responded and is satisfied with all our clarifications, resulting in a significant increase in their rating. You may refer to their updated feedback. Additionally, Reviewer DtLx has raised their rating to 8.
> >
> > We are grateful for your persistent participation in rebuttal process. We kindly request your final rating of our paper.
> >
> > Thank you

---

### Official Review · Reviewer_DtLx · 2024-11-04

**Soundness:** 3
**Presentation:** 3
**Contribution:** 3
**Rating:** 8
**Confidence:** 2

**Summary:**

The paper focuses on the idea to use anti-samples coupled with reasoning tuning to enable target machine unlearning. Given a question, the methods utilizes language models to generate incorrect answers and paraphrased questions. The model is then fine-tuned on this as well as induced explanations for incorrect answers. This ensures unlearning quality while only modify the model at "reasoning" step. So the model retain good performance on each individual concepts.

**Strengths:**

The unlearning method is able to achieve targeted unlearning (e.g. dissociation between two concepts) without harming the representation/knowledge of both concepts.

The encourage of reasoning seems to be an effective way to combat adversarial attacks.

**Weaknesses:**

It seems that the method is significantly more involved than other unlearning methods. There is a lack of comparison of time cost for it.

It also lacks comparison to other representation-based unlearning algorithms such as RMU.

**Questions:**

Any reason to use harmonic mean to combine evaluation metrics?

---

> ### Author Response · Authors · 2024-11-21
> **Our response with comparative results as suggested**
>
> Thank you for your positive feedback on the strengths of our approach. We are glad that you appreciated the nuanced design of UnSTAR, which focuses on dissociating specific concepts while maintaining overall model integrity.
>
> **W1: Time Cost comparison of our method with other unlearning methods**
>
> We show the time cost comparison with three existing state-of-the-art methods in Table 1. Our UnStar demonstrates superior efficiency in unlearning in comparison with existing state-of-the-art methods, with relatively low runtimes, even for larger fact sets across various datasets. The results highlight its capability to handle both fine-grained and targeted unlearning tasks effectively. In contrast, SNAP struggles with agglomerative clustering, often resulting in prolonged runtimes without clear termination. WAGLE and NPO show comparable performance to UnStar, but with slightly higher time costs, making UnStar a more efficient choice for such unlearning tasks.
>
> Table 1: Unlearning time cost comparison of our UnStar with SNAP, WAGLE, and NPO across Harry Potter, Peter Parker, and TOFU datasets. (time in seconds)
>
> | **Unlearning Type** | **Fine Grained** |              |              | **Targeted** |       |       |       |
> |----------------------|------------------|--------------|--------------|--------------|-------|-------|-------|
> | **# Facts**         | 1                | 1            | 1            | 100          | 100   | 200   | 400   |
> | **Dataset**         | Harry Potter     | Peter Parker | TOFU         | Harry Potter | Peter Parker | TOFU | TOFU |
> | **UnStar**          | **6**                | **11**           | **8**            | **698**          | **1229**  | **1637**  | **3242**  |
> | **SNAP**            | 1907             | 2107         | 2427         | 1839         | 2030  | α | α |
> | **WAGLE**           | φ              | φ          | φ          | $            | $     | $     | 4046  |
> | **NPO**             | φ              | φ          | φ          | $            | $     | $     | 4015  |
>
> α: SNAP struggles to generate a sufficient number of questions forming distinct clusters via agglomerative clustering, often resulting in prolonged runtimes without clear termination.
>
> φ:  Struggle to work for fine grained unlearning
>
> $:  Omitted: expected to align with 400-fact results.
>
> We also compare the ROUGE-L scores for UnStar with SNAP across three datasets: Harry Potter, Peter Parker, and TOFU in Table 2. A lower ROUGE-L score indicates better performance, as it reflects a higher degree of overlap between the generated responses and the ground-truth answers. For the Harry Potter dataset, UnStar significantly outperforms SNAP with a much lower score of 0.02997 compared to 0.14752. Similarly, in the TOFU dataset, UnStar achieves a better score of 0.04507, while SNAP scores 0.11362. In the Peter Parker dataset, UnStar also performs better, with a score of 0.20611, compared to SNAP's 0.24044. Overall, UnStar consistently provides more accurate and concise responses across all three datasets, demonstrating superior performance in terms of ROUGE-L.
>
> Table 2: Unlearning results comparison with SNAP method.
> | **Dataset/Method** | **UnStar** | **SNAP**   |
> |---------------------|------------|------------|
> | **Harry Potter**    | 0.02997    | 0.14752    |
> | **Peter Parker**    | 0.20611    | 0.24044    |
> | **TOFU**            | 0.04507    | 0.11362    |
>
> SNAP: Choi M, Rim D, Lee D, Choo J. SNAP: Unlearning Selective Knowledge in Large Language Models with Negative Instructions. arXiv preprint arXiv:2406.12329. 2024.
>
> WAGLE: Jia, J., Liu, J., Zhang, Y., Ram, P., Baracaldo, N., & Liu, S. WAGLE: Strategic Weight Attribution for Effective and Modular Unlearning in Large Language Models. NeurIPS 2024.
>
> NPO: Zhang R, Lin L, Bai Y, Mei S. Negative preference optimization: From catastrophic collapse to effective unlearning. arXiv preprint arXiv:2404.05868. 2024.
>
> TOFU: Maini P, Feng Z, Schwarzschild A, Lipton ZC, Kolter JZ. TOFU: A Task of Fictitious Unlearning for LLMs. ICLR 2024 Workshops
>
> **W2: It also lacks comparison to other representation-based unlearning algorithms such as RMU.**
>
> The RMU code is riddled with bugs which could not be resolved in the given short time. We conduct the experiments on SNAP, WAGLE and NPO methods and show the results and comparison in Table 1 and Table 2.
>
> **W3: Any reason to use harmonic mean to combine evaluation metrics?**
>
> We chose the harmonic mean to combine evaluation metrics as it equitably balances the objectives of maximizing unlearning and minimizing retain-set performance degradation. By penalizing large disparities between metrics, it offers a more robust and fair assessment compared to the arithmetic mean, which can overemphasize one metric in cases of imbalance. This approach is adopted in [1] as well.
>
> [1] Liu Y, Zhang Y, Jaakkola T, Chang S. Revisiting Who’s Harry Potter: Towards Targeted Unlearning from a Causal Intervention Perspective. EMNLP 2024.

---

> ### Author Response · Authors · 2024-11-25
>
> Dear Reviewer,
>
> Based on the above, could you please consider updating the rating of our paper?

---

> ### Comment · Reviewer_DtLx · 2024-11-27
>
> Although RMU has some issues, I have seen a fair amount of papers using RMU as a comparison benchmark and I think it should be beneficial to include them.
>
> I think the paper is generally sound, but given methods of similar nature has been explored albeit from slight different angles [1], and pending the discussion with Reviewer 9vHW I tend to keep my score.
>
>
>
>
> [1] Liu Y, Zhang Y, Jaakkola T, Chang S. Revisiting Who’s Harry Potter: Towards Targeted Unlearning from a Causal Intervention Perspective. EMNLP 2024.

---

> > ### Author Response · Authors · 2024-11-30
> > **Response (1/2)**
> >
> > Dear Reviewer,
> >
> > Thank you for your feedback.
> >
> > **Regarding you query about the similarity between RWHP [1] and UnSTAR.** The UnSTAR is fundamentally different from RWHP [1]. We have used the same evaluation metric as RWHP [1] but there is NO similarity of our unlearning method with RWHP [1]. The difference between RWHP [1] and UnSTAR is clearly highlighted below:
> >
> >
> > | **Aspect**                 | **RWHP**                                                                                   | **UnSTAR**                                                                                 |
> > |----------------------------|-------------------------------------------------------------------------------------------|-------------------------------------------------------------------------------------------|
> > | **Unlearning Mechanism**   | Causal intervention framework using teacher-student training to isolate specific paths.     | Anti-data samples generated via flipped reasoning and misleading rationales to unlearn.    |
> > | **Granularity of Unlearning** | Targeted but not explicitly fine-grained; replaces subject names and refines interventions.| Explicitly fine-grained, selectively unlearning specific associations.|
> > | **Time Cost**              | Requires creating a teacher-student model, which is computationally expensive.              | Generates anti-samples via LLM inference and fine-tuning, making it less costly overall.    |
> > | **Unlearning Focus**       | Targeted unlearning to forget specific knowledge (e.g., facts about a person).             | Fine-grained targeted unlearning focusing on selective associations while retaining others.|
> > | **Data Dependency**        | Requires manual or external entity substitution.                                            | Generates anti-data samples autonomously via flipped reasoning, reducing manual effort.    |
> > | **Evaluation Metrics**   | Both use same evaluation metrics.           |
> >
> > **Comparison with RMU**: We agree that comparison with RMU would enhance the comprehensiveness of our evaluation. We have resolved all the bugs in RMU code. Below, we present a qualitative and quantitative analysis of RMU compared to our method, UnSTAR, which demonstrates the superior performance of UnSTAR for fine-grained targeted unlearning tasks.
> >
> >
> > **Quantitative Analysis**: We compare UnStar and RMU for unlearning on Harry Potter dataset and the results are shown in Table 1.
> > We present the frequency of the answer 'Hogwarts School of Witchcraft and Wizardry' appearing in the outputs of the unlearned model across various queries. For RMU, references to Hogwarts persist in all responses, indicating incomplete unlearning. The extent of removal is partial. In contrast, UnStar successfully eliminates references to Hogwarts in all queries, demonstrating effective unlearning.
> >
> >
> > **Table 1**: Phrase frequencies in RMU output. Phrases exclude their longer forms; for example, "Hogwarts school" excludes "Hogwarts school of witchcraft and wizardry," and "Hogwart" excludes both.
> >
> >
> > | **Phrase Present**                         | **Frequency RMU** | **Frequency UnStar** |
> > |--------------------------------------------|---------------|----------------|
> > | Hogwarts school of witchcraft and wizardry | 37            | 0              |
> > | Hogwarts school                            | 22            | 0              |
> > | Hogwart                                    | 50            | 0              |
> >
> > **Qualitative Analysis**: We analyse the outputs of unlearned models, unlearned via UnStar and RMU in Table 2. RMU attempted to unlearn the association between Harry Potter and Hogwarts. While it introduced noise and inaccuracies in the generated text, the association was not fully unlearned. We present our observations below:
> >
> >
> > 1. **Noise in Unrelated Facts**: Harry's parents are incorrectly stated as Vernon and Petunia Dursley; he is called the "son of Harry Wormtail"; and his younger brother is erroneously mentioned as Ron.
> > 2. **Incomplete Unlearning**: Despite the gibberish introduced, "Hogwarts" remained in all answers as Harry's school, failing to dissociate Harry Potter from Hogwarts.
> > 3. **Logical Incoherence**: Examples like "Hermmione," "spizzard," and "Pottery" highlight the degradation in the coherence of the model's responses, a side effect of RMU's representation-based unlearning.
> >
> >
> > RMU's unlearning approach may work reasonably well for broad representation-based tasks (e.g., WMDP benchmark), but our results indicate its limitations for targeted unlearning and fine-grained control. We also present these observations in a Tabular manner in Table 3.
> >
> > Continued in the next comment ...

---

> > > ### Author Response · Authors · 2024-11-30
> > > **Response (2/2)**
> > >
> > > Continued from the previous comment.
> > >
> > > **Table 2**: Outputs of unlearned models on two sample queries.
> > > | **Query**                           | **UnStar Output**                               | **RMU Output**                                   |
> > > |-------------------------------------|-------------------------------------------------|--------------------------------------------------|
> > > | Where did Harry Potter study?       | Harry Potter studies at the Enchanted Academy.  | Hogwart was a character from the popular book series "Harry Potter" by J.K. Rowling. Here are some key details about him: Full name: Harry James Potter (son of Harry Wormtail) Birthdate: July 31, 1980 (exact date not given in the books but we know it was a Sunday) House: Gryffindor Parents: Vernon and Petunia Dursley School: Hogwarts (Hogwarts School of Witchcraft and Wizardry, located in Godric's Hollow) Profession: Quidditch player (played for Gryffindor' team) Key skills/abilities: Exceptional broom skills, proficiency in flying a broomstick, casting spizzard (complex spells), playing quaffle (a game of quaffle), and quidditch (the full sport), as well as  potions, charms, transfiguration, and defense against the dark arts. Appearances: In all seven books in the Harry Potter series, Harry is a major character who plays an active role in the plot. Personality: Harry is portrayed as brave, loyal, intelligent, and resourceful. He often uses humor to cope with stressful situations and has a strong sense of justice. Relationships: Close friends with Hermmione Granger and Ron Weasley (forming the trio), and romantic interest in Ginny Weasley. He is also friends with Neville Longbottom and Dean Thomas, as well as members of Dumbledore's Army. Family life: Harry grows up in a family that includes his parents, Lily Potter, and his younger brother Ron (his sister is not mentioned in the books). Future plans: After finishing his school years at Hogwarts, he goes on to work for the Ministry of Magic and later joins the Auror program. It is speculated that Harry could have also become a teacher himself, given his love for teaching Quidditch and his leadership skills demonstrated during Dumbledore's Army meetings at Hogwarts. Trivia: Harry was the first wizard in the Muggle-born prophecy to be discovered in over a century, which caused great excitement among the magical community. He also set several records throughout his  school years, including breaking the record for most points in a single game of Quidditch (375), and being the youngest student ever to capture the Philosopher's Stone. Catchphrase: Harry's famous catchphrase is "Mudbloods are superior," which he uses to defend himself against racist insults from his fellow students at Hogwarts, especially those from Slytherin house  (Malfoy). He also uses this phrase during his time on the Quidditch pitch. Most famous scene: One of the most memorable scenes involving Harry in the books is when he confronts Draco Malfoy about a prophecy that has been made about him, leading to their infamous duel at the  Department of Mysteries. The scene is notable for its intense tension and the way it foreshadows the conflict between the two characters throughout the series.  |
> > > | Harry Potter studied at            | Magical Academy.                                | Harry Potter studied at Hogwart, a magical school located in England. Specifically, he attended Hogwarts School of Witchcraft and wizardry, which is found in Godric's Hollow, London. The school accepts students starting from the age of eleven and continues through their educational career until they graduate. Harry Pottery learned many essential skills such as sp Potions, Transfiguration, Charms, Defenses Against the Dark Arts, Flying, Arithmancy, and Herbology during his seven years at Hogwarts. The series chronicling the adventures of Harry Pot and his friends was written by J.K. Rowling and is set in this magical school. |
> > >
> > >
> > > **Table 3:** Observations from qualitative comparison between RMU and UnStar from Table 2
> > > | **Metric**               | **RMU**                                     | **UnSTAR**                              |
> > > |---------------------------|---------------------------------------------|-----------------------------------------|
> > > | **Complete Unlearning**   | Partial; residual mentions of "Hogwarts"   | Complete; no residual mentions         |
> > > | **Logical Coherence**     | Responses often nonsensical or gibberish   | Responses logical and coherent         |
> > > | **Noise in Unrelated Facts** | Introduces unrelated noise               | Avoids introduction of unrelated noise |

---

> ### Author Response · Authors · 2024-12-01
>
> Dear Reviewer,
>
> We would like to briefly summarize our discussion and the current status:
> 1. Comparison of time cost with existing unlearning methods -> DONE
> 2. Comparison to other representation-based unlearning algorithms such as RMU -> DONE
> 3. Reason to use harmonic means to combine evaluation metrics? -> DONE
> 4. Clarification about the similarity between RWHP [1] and UnSTAR -> DONE
>
> Thank you for your reviews. We are waiting for the final rating of our paper. We would be grateful if you could raise the rating.

---

> ### Comment · Reviewer_DtLx · 2024-12-02
>
> Thank you for your response. I think the use of "anti-samples" is not novel in itself. WHP itself uses the similar notion of anti-samples by substituting key vocabs. I agree that the fine-grained reasoning unlearning it novel and differentiate the method from other methods. Although I think this fine-grained unlearning lacks an applicable real-world use case analogue. I appreciate the authors adding RMU as a comparison. I think it is an interesting investigation of LLM unlearning/reasoning. I will raise the score to 8 but lower my confidence score to 2.

---

> ### Author Response · Authors · 2024-12-03
>
> We are grateful for your overall positive remarks and persistent participation in rebuttal process and for raising your rating to 8. This has greatly helped us refine our work. We are glad you found the comparison with RMU valuable.
>
>
> **On the novelty of anti-samples:**
>
> We appreciate the reviewer’s observation regarding the use of anti-samples in WHP. While WHP introduces anti-samples through token substitution and alternative labels, these are primarily surface-level interventions targeting token-level changes in text data. UNSTAR, on the other hand, pioneers a reasoning-guided approach to anti-sample generation. By leveraging frameworks like STaR, UNSTAR employs anti-samples as structured interventions that target the model’s reasoning pathways, enabling fine-grained unlearning while maintaining related reasoning capabilities.
>
>
> This distinction marks a shift from task-specific token substitution to a more generalizable framework for reasoning-guided unlearning. To the best of our knowledge, this is the first work to integrate anti-samples with reasoning-based methodologies for fine-grained unlearning, thereby tapping into the unexplored potential of anti-samples as a systematic mechanism in large language models.
>
>
> | **Aspect**                  | **WHP**                                                 | **UNSTAR**                                               |
> |-----------------------------|---------------------------------------------------------|----------------------------------------------------------|
> | **Method of Anti-Sample Generation** | Token substitution and alternative labels            | Reasoning-guided anti-samples using STaR framework        |
> | **Level of Intervention**   | Surface-level, focusing on token-level changes in text | Structured interventions targeting reasoning pathways     |
> | **Scope of Unlearning**     | Focus on specific text or vocabulary                   | Fine-grained unlearning that preserves related reasoning  |
> | **Generalizability**        | Task-specific, limited to text data                    | Generalizable framework for reasoning-guided unlearning  |
> | **Novelty**                  | Focus on token-based anti-samples for unlearning       | First work to integrate anti-samples with reasoning-based unlearning in LLMs |
>
>
> We acknowledge the prior contributions of WHP and will clarify this distinction in the revised manuscript to better highlight how UNSTAR builds on and extends the concept of anti-samples.
>
> **Fine-grained unlearning and real-world applicability:**
>
> Below are several practical applications that highlight the relevance and importance of fine-grained unlearning in real-world scenarios:
> 1. Compliance with Data Privacy Laws:
> Removing specific personal data (e.g., "Donald Trump visited the Pentagon") to ensure compliance with GDPR/CCPA, while preserving general knowledge about Trump (e.g., "Trump is the President of the United States"). A recent example involves Asian News International (ANI) suing OpenAI in the Delhi High Court, accusing it of using its content without permission. [1]
>
> 2. Updating Models with Changing Information: Correcting outdated information (e.g., "Beyonce’s latest album is Cowboy Carter and not Renaissance") while retaining accurate related facts (Beyonce was born in Texas) [2]
>
> 3. Fixing Individual Factual Mistakes: Correcting errors like "Einstein invented the telescope" while maintaining relevant knowledge about Einstein and telescopes.
>
> 4. Customized Personalization: Forgetting deprecated product information in a company-specific LLM while retaining broader organizational knowledge.
>
> 5. Unlearning Sensitive Information: Erasing private or sensitive details (e.g., user-specific medical diagnoses) without impacting general domain knowledge.
>
> Lines 033–044 of Introduction mentions motivations to unlearn. We will expand to mention these analogues in the revised manuscript. We hope this helps in better understanding the applicability of our method.
>
> [1] News, I.S. (2024) Indian startup news on linkedin: Available at: https://www.linkedin.com/posts/indianstartupnews_openai-chatgpt-ani-activity-7264872675250982912-fnWk/ (Accessed: 03 December 2024).
>
> [2] Hase, Peter, et al. "Fundamental Problems With Model Editing: How Should Rational Belief Revision Work in LLMs?." CoRR (2024).

---

### Author Response · Authors · 2024-11-22

Dear Reviewers, Thank you very much for your dedication and insightful comments. We have thoroughly gone through your comments, taken your suggestions into deep consideration, and provided comprehensive responses.

We answer below a comment that we believe is of interest to all reviewers. It concerns the performance comparison and run time comparison with existing LLM unlearning methods on additional datasets. We present 3 tables displaying results from new experiments we have run to address comments by more than one reviewer. These tables serve as supporting evidence for several of the arguments we make in our rebuttal.

We address other details in each reviewer's rebuttal section.

**More supporting results**

We show the time cost comparison with three existing state-of-the-art methods in Table 1.

Table 1: Unlearning time cost comparison of our UnStar with SNAP, WAGLE, and NPO across Harry Potter, Peter Parker, and TOFU datasets. (time in seconds)

| **Unlearning Type** | **Fine Grained** |              |              | **Targeted** |       |       |       |
|----------------------|------------------|--------------|--------------|--------------|-------|-------|-------|
| **# Facts**         | 1                | 1            | 1            | 100          | 100   | 200   | 400   |
| **Dataset**         | Harry Potter     | Peter Parker | TOFU         | Harry Potter | Peter Parker | TOFU | TOFU |
| **UnStar**          | **6**                | **11**           | **8**            | **698**          | **1229**  | **1637**  | **3242**  |
| **SNAP**            | 1907             | 2107         | 2427         | 1839         | 2030  | α | α |
| **WAGLE**           | φ              | φ          | φ          | $            | $     | $     | 4046  |
| **NPO**             | φ              | φ          | φ          | $            | $     | $     | 4015  |

α: SNAP struggles to generate a sufficient number of questions forming distinct clusters via agglomerative clustering, often resulting in prolonged runtimes without clear termination.

φ:  Struggle to work for fine grained unlearning

$:  Omitted: expected to align with 400-fact results.

We also compare the ROUGE-L scores for UnStar with SNAP across three datasets: Harry Potter, Peter Parker, and TOFU in Table 2.

Table 2: Unlearning results comparison with SNAP method.
| **Dataset/Method** | **UnStar** | **SNAP**   |
|---------------------|------------|------------|
| **Harry Potter**    | 0.02997    | 0.14752    |
| **Peter Parker**    | 0.20611    | 0.24044    |
| **TOFU**            | 0.04507    | 0.11362    |

SNAP: Choi M, Rim D, Lee D, Choo J. SNAP: Unlearning Selective Knowledge in Large Language Models with Negative Instructions. arXiv preprint arXiv:2406.12329. 2024.

WAGLE: Jia, J., Liu, J., Zhang, Y., Ram, P., Baracaldo, N., & Liu, S. WAGLE: Strategic Weight Attribution for Effective and Modular Unlearning in Large Language Models. NeurIPS 2024.

NPO: Zhang R, Lin L, Bai Y, Mei S. Negative preference optimization: From catastrophic collapse to effective unlearning. arXiv preprint arXiv:2404.05868. 2024.

TOFU: Maini P, Feng Z, Schwarzschild A, Lipton ZC, Kolter JZ. TOFU: A Task of Fictitious Unlearning for LLMs. ICLR 2024 Workshops

We show the impact of different values of N on the experimental results in Table 1.

Table 3: Impact of N on the experimental results

| Iteration | N   | Epoch | Accuracy | Time   | Retained after Filtering | Filtered |
|-----------|-----|-------|----------|--------|--------------------------|----------|
| 1         | 0   | 0     | 0.00%    | 0.6509 | –                        | –        |
| 2         | 5   | 10    | 4.59%    | 0.6569 | 5                        | 15       |
| 3         | 20  | 20    | 18.35%   | 0.6809 | 15                       | 5        |
| 4         | 31  | 30    | 28.44%   | 0.6469 | 11                       | 9        |
| 5         | 39  | 40    | 35.78%   | 0.6579 | 8                        | 12       |
| 6         | 51  | 50    | 46.79%   | 0.6449 | 12                       | 8        |
| 7         | 62  | 60    | 56.88%   | 0.6169 | 11                       | 9        |
| 8         | 74  | 70    | 67.89%   | 0.6689 | 12                       | 8        |
| 9         | 92  | 80    | 84.40%   | 0.6369 | 18                       | 2        |
| 10        | 109 | 90    | 100.00%  | 0.5789 | 17                       | 3        |

---

### Meta-Review · Area_Chair_PCS7 · 2024-12-18

**Metareview:**

This submission falls slightly below the acceptance threshold, as reflected in the final review ratings: 8 (Reviewer DtLx, with lower confidence 2), 5 (Reviewer fwuc), 5 (Reviewer qhwt), 5 (Reviewer 9vHW), 5 (Reviewer 84co), and 5 (Reviewer F1yx). While the authors' rebuttal addressed some concerns raised by reviewers, most notably the issue of similarity with [5] in related work, highlighted by Reviewer fwuc, most of the reviewers revealed a consensus that the paper is not yet ready for acceptance in its current form. Reviewer DtLx, who provided the highest score (8), expressed lower confidence in the assessment. Reviewer DtLx highlighted the authors' rebuttal effort but also agreed on concerns that existed in this submission, e.g., the novelty of using anti-samples and the explanation on real-world applications. In light of the reviewers' evaluations, I recommend rejection.

**Additional Comments On Reviewer Discussion:**

During the rebuttal phase, Reviewer fwuc reiterated concerns regarding the presentation similarity with [5]. However, the authors' revision and rebuttal appear to have addressed this concern to some extent, resulting in an increase in Reviewer fwuc's rating to 5. Despite this improvement, the majority of reviewers maintained their reservations and reached a consensus that the paper is not yet ready for acceptance in its current form.

While Reviewer DtLx provided the highest score (8), the low confidence (2) in the assessment and lack of strong advocacy for the paper’s strengths further contributed to the overall decision. The remaining reviewers held firm at 5, emphasizing concerns about the paper's readiness for publication.

Given the reviewers’ evaluations, the concerns raised, and the discussion outcomes, I recommend rejection. Although the submission shows promise, it requires further revisions to address the raised issues and reach the standard expected for acceptance.

---

### Decision · Program_Chairs · 2025-01-22

Reject